# Foreground-Aware Token Routing Vision Transformer for Real-Time Satellite Video Tracking

**Jiahao Wang**[1] **Fang Liu**[1] **Licheng Jiao**[1] **Shuo Li**[1]
**Hao Wang**[1] **Lingling Li**[1] **Xinyi Wang**[1] **Xu Liu**[1]

## Abstract

Real-time satellite video tracking poses distinct challenges, including accommodating high spatial-temporal resolution, dynamic backgrounds, and constrained onboard computational resources. While Discriminative Correlation Filter (DCF)-based methods offer high-speed inference, they suffer from limited accuracy. In contrast, Vision Transformer (ViT)-based trackers achieve strong performance by unifying representation and aggregation in a single-stream design, yet their heavy computational footprint limits practical deployment in real-time satellite scenarios. In this work, we present FATrack, a novel tracking framework that effectively balances tracking accuracy and computational efficiency. At its core is FA-ViT, a lightweight Vision Transformer backbone that introduces foreground-aware token routing, enabling the model to concentrate computation on target-relevant regions while suppressing redundancy. To mitigate semantic degradation caused by token sparsification, we propose the Adaptive Scatter Module (ASM), which selectively reinforces informative tokens via joint spatial-channel attention and sparse structural propagation, thereby enhancing semantic fidelity and spatial coherence. By synergistically integrating FA-ViT and ASM, FATrack forms a unified architecture that delivers real-time performance with significantly improved tracking precision. Extensive evaluations on multiple satellite video benchmarks demonstrate that FATrack outperforms existing real-time trackers while achieving DCF-level inference efficiency, showing strong potential for practical large-scale satellite video tracking.

[1]School of Artificial Intelligent, Xidian University, Xi'an,710071, P.R. China. Correspondence to: Fang Liu <jh_wang1024@163.com, f63liu@163.com>.

*Proceedings of the 43rd International Conference on Machine Learning*, Seoul, South Korea. PMLR 306, 2026. Copyright 2026 by the author(s).

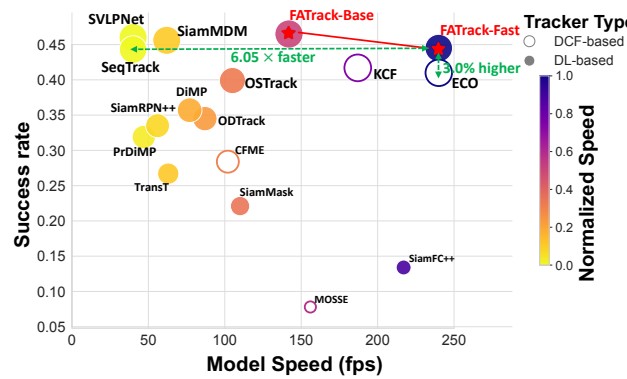

*Figure 1.* **A speed-accuracy trade-off evaluation is conducted on the SV248S.** FATrack achieves higher accuracy compared to traditional trackers with similar speed (e.g., a 3% improvement over ECO (Danelljan et al., 2017)), while maintaining comparable performance at significantly higher speed (e.g., 6.05× faster than SeqTrack (Chen et al., 2023)).

## 1. Introduction

Satellite video tracking has become a cornerstone task in remote sensing applications such as urban monitoring (Albert et al., 2017; Zhao et al., 2017), maritime surveillance (Szpak & Tapamo, 2011), and disaster response (Barnes et al., 2007; Liu & Hodgson, 2016). The task demands high tracking accuracy under complex conditions, such as low object resolution, scale variations, cluttered backgrounds, and strict real-time performance for mission-critical scenarios (Jiao et al., 2019; Shao et al., 2019b; Wang et al., 2020b; Zhang et al., 2021). This dual requirement places significant constraints on the design of effective tracking algorithms in the satellite domain.

Discriminative Correlation Filter (DCF)-based trackers (Bolme et al., 2010; Danelljan et al., 2017; Pang et al., 2023) have long dominated satellite video tracking due to their lightweight design and low computational overhead. However, their limited modelling capacity and reliance on handcrafted features hinder their tracking robustness and accuracy, especially in highly dynamic environments. In contrast, recent advances in deep learning (DL)-based trackers (Nam & Han, 2016; Guo et al., 2021; Xie et al., 2024), particularly those leveraging Vision Transformers (ViTs) (Chen et al., 2023; Wang et al., 2024b), have demonstrated

remarkable improvements in precision by unifying feature extraction and fusion within a powerful end-to-end framework. Nevertheless, these models often suffer from substantial latency and heavy computational costs, rendering them unsuitable for real-time tracking in satellite scenarios.

Despite recent advances in satellite video tracking (Shao et al., 2019a; Jiao et al., 2023; Shao et al., 2021; Yang et al., 2023a), a fundamental tension remains unresolved: existing ViT-based trackers (Wang et al., 2024a; Zheng et al., 2024) excel in accuracy but suffer from high latency, while traditional DCF-based methods offer real-time performance at the cost of poor discriminability. This dichotomy arises from an architectural gap: Generic transformer designs prioritize global representation without regard to computational locality or target-specific focus, making them ill-suited for satellite videos characterized by low temporal resolution, sparse motion cues, and expansive spatial contexts. To bridge this gap, we argue for a task-driven rethinking of ViT (Dosovitskiy et al., 2020) that aligns representation learning with the operational demands of satellite tracking. Specifically, we advocate a foreground-aware, sparsity-enhanced formulation that allocates computational resources to semantically relevant regions while preserving the expressive capacity of transformers, enabling fast and reliable tracking in real-world aerial video streams.

To this end, we propose FATrack, a Foreground-Aware Token Routing Vision Transformer that strikes an optimal balance between high-precision tracking and real-time efficiency. At the core of FATrack lies FA-ViT, a lightweight transformer backbone that redefines token routing through the lens of foreground awareness, prioritizing semantically relevant and spatially focused regions while suppressing redundant computation in background areas. To address the critical challenge of semantic degradation caused by token sparsification, we further introduce the Adaptive Scatter Module (ASM). It selectively propagates semantic cues from informative foreground tokens to underrepresented spatial locations, thus enhancing structural continuity and target fidelity in the reconstructed feature space. By jointly optimizing for computational efficiency, spatial discrimination, and semantic consistency, FATrack moves beyond the traditional accuracy-speed trade-off. Delivers real-time performance comparable to DCF-based trackers while achieving significantly higher tracking precision. Extensive experiments on standard satellite tracking benchmarks demonstrate the superior accuracy-speed trade-off of our approach, setting a new standard for practical satellite video tracking. Figure 1 shows the superiority of FATrack, which has favorable speed-precision trade-offs. Our contributions can be summarized as follows:

- We propose FA-ViT, a lightweight yet expressive Vision Transformer backbone specifically tailored for

real-time satellite video tracking characteristics. FA-ViT reconfigures token interaction patterns to prioritize foreground modelling and computational locality, achieving high efficiency without compromising representational capacity.

- Built upon FA-ViT, we develop FATrack. This real-time and accurate tracker bridges the gap between efficiency-driven DCF-based approaches and accuracy-focused deep learning-based methods. FATrack inherits the precision benefits of Transformer architectures while operating at speeds comparable to lightweight DCF trackers.

- We introduce the Adaptive Scatter Module (ASM) to enable fine-grained, foreground-aware token enhancement through joint spatial-channel attention and sparse update operations. ASM adaptively routes salient information into the extended representation space, boosting target localization accuracy with negligible overhead.

- Extensive experiments on multiple satellite video benchmarks validate the effectiveness of FATrack, showing consistent improvements over both lightweight DCF-based baselines and heavyweight ViT-based trackers in terms of accuracy and speed.

## 2. Related Work

**Satellite Video Tracking.** Satellite video tracking has emerged as a crucial task for high-resolution remote sensing applications, including urban monitoring, environmental surveillance, and disaster response. Due to the wide-area coverage, low frame rate, and complex motion patterns in satellite videos, designing robust and efficient tracking algorithms remains a significant challenge. Traditional approaches are predominantly dominated by Discriminative Correlation Filter (DCF)-based methods (Bolme et al., 2010; Henriques et al., 2012; 2014a; Danelljan et al., 2016; 2017; Kiani Galoogahi et al., 2017a; Lukezic et al., 2017; Li et al., 2018b; Bhat et al., 2019; Danelljan et al., 2020; Zheng et al., 2020) due to their lightweight design and real-time performance. Representative works such as CFME (Xuan et al., 2019) and CPKF (Li et al., 2022a) have explored template matching and correlation mechanisms to adapt DCF paradigms to satellite data. However, the limited representation capacity of handcrafted features or shallow backbones restricts their tracking precision, especially under cluttered backgrounds or abrupt motion. More recently, deep learning-based methods have shown promise in satellite video tracking. These approaches often leverage pre-trained convolutional networks or Transformer-based architectures to enhance feature expressiveness and improve tracking robustness. For instance, adaptations of Siamese networks (Shao et al., 2019a; 2021; Yang et al., 2023a;b) and Transformer-

based frameworks (Chen et al., 2023; Wang et al., 2024a;c; Xie et al., 2024; Wang et al., 2024b; Zheng et al., 2024) have demonstrated improved accuracy. Nonetheless, such architectures often incur high computational costs, making them impractical for deployment in latency-sensitive or large-scale satellite video analytics. Thus, achieving an optimal balance between tracking accuracy and inference efficiency remains a significant challenge in this domain.

**Efficient Vision Transformer.** Vision Transformers (ViTs) (Dosovitskiy et al., 2020; Han et al., 2022) have significantly advanced visual representation learning by leveraging global context modelling through self-attention mechanisms. However, the inherent quadratic complexity of vanilla self-attention for input tokens presents substantial computational burdens, making standard ViTs ill-suited for real-time or resource-constrained environments. To mitigate this, a range of efficient Vision Transformer variants have been introduced (Rao et al., 2021; Meng et al., 2022; Yin et al., 2022; Tang et al., 2022b; Liu et al., 2023; Papa et al., 2024). Notably, architectures such as MobileViT (Mehta & Rastegari, 2021), LeViT (Graham et al., 2021), and Efficient-Former (Li et al., 2022c) incorporate hybrid convolution-attention designs or low-rank factorization strategies to reduce computational overhead while maintaining expressive capacity. Similarly, sparse attention mechanisms (Ren et al., 2021; Chen et al., 2021a) (e.g., Performer (Choromanski et al., 2020), Linformer (Wang et al., 2020a)) and hierarchical/local attention structures (e.g., Swin Transformer (Liu et al., 2021)) aim to enhance scalability on high-resolution inputs. Despite their success in image classification and general vision tasks, these designs are not directly applicable to satellite video tracking, which imposes unique challenges such as low frame rates, wide-area scenes, sparse motion patterns, and cluttered backgrounds. These characteristics demand temporal stability and discriminative focus, and highly efficient computation. Existing lightweight ViTs (Zheng et al., 2023; Fan et al., 2023; Tang et al., 2022a; Luo et al., 2022; Qian et al., 2023) often lack task-specific inductive biases, such as token prioritization and foreground-background decoupling, that are essential for robust tracking under satellite-specific constraints. Consequently, a purpose-built Vision Transformer architecture is needed to bridge the gap between efficiency and discriminative tracking performance in satellite video scenarios.

To address these, we introduce FA-ViT, a lightweight Vision Transformer architecture specifically tailored for satellite video tracking. By incorporating foreground-aware token routing and enforcing computation locality, FA-ViT strikes a fine balance between representational expressiveness and computational efficiency. This design enables effective modelling of object-centric dynamics in satellite scenarios while maintaining real-time performance.

## 3. Methodology

### 3.1. Overall

The FATrack is proposed to address the challenges of redundant computation and foreground-background ambiguity in satellite video tracking. Instead of uniformly processing all spatial tokens, FATrack prioritises foreground semantics while suppressing irrelevant backgrounds, enabling more efficient and robust tracking in cluttered remote sensing scenarios. The overall structure is shown in Figure 2. At the heart of FATrack lies a Foreground-Aware Token Routing Vision Transformer (FA-ViT), which follows a U-Net-inspired (Ronneberger et al., 2015) spatial shrinkage expansion strategy. Specifically, early transformer layers progressively shrink the active spatial region, focusing computation toward a center-aligned token region presumed to contain the target. Later layers recover the spatial extent, guided by learned semantic dependencies. An Adaptive Scattering Module (ASM) is introduced to enhance this process and generate meaningful representations for newly expanded tokens by diffusing semantic information from the retained core tokens. The overall process can be formalized as:

$$
\boldsymbol{x}^{(i+1)} = \begin{cases} \text{Shrink}\left(\boldsymbol{x}^{(i)}, \boldsymbol{r}_i\right), & i < L/2 \\ \text{Expand}\left(\boldsymbol{x}^{(i)}, \boldsymbol{r}_i, \text{ASM}\right), & i \geq L/2 \end{cases} \quad (1)
$$

where $\boldsymbol{x}^{(i)}$ denotes the token representation at the $i$-th transformer block, and $\boldsymbol{r}_i$ is the spatial resolution at that stage. **Shrinkage** reduces spatial extent while preserving semantics; **Expansion** restores resolution while maintaining semantic fidelity via the ASM. By jointly optimizing token routing and semantic reconstruction, FATrack enables the model to operate lightweight and target-focused, significantly improving tracking accuracy and efficiency in complex satellite video scenes.

### 3.2. FA-ViT

To achieve efficient and accurate tracking in real-time satellite video, we propose the Foreground-Aware Token Routing Vision Transformer (FA-ViT), a lightweight and spatially adaptive ViT backbone tailored for online tracking scenarios. The motivation behind FA-ViT is twofold: (1) targets in satellite video are typically centered in the search region due to motion prediction and search window strategies; (2) the surrounding background regions often introduce redundant computation and degrade tracking robustness. Therefore, we introduce a novel token routing mechanism that dynamically adjusts the spatial token field throughout the transformer layers, focusing attention on central foreground regions and suppressing irrelevant context. Given a template image $\boldsymbol{Z} \in \mathbb{R}^{H \times W \times 3}$ and a search image $\boldsymbol{X} \in \mathbb{R}^{H \times W \times 3}$, we apply a shared patch embedding $\phi(\cdot)$, obtaining tokens $\boldsymbol{T}_Z = \phi(\boldsymbol{Z}) \in \mathbb{R}^{N_Z \times d}$ and $\boldsymbol{T}_X = \phi(\boldsymbol{X}) \in \mathbb{R}^{N_X \times d}$, with

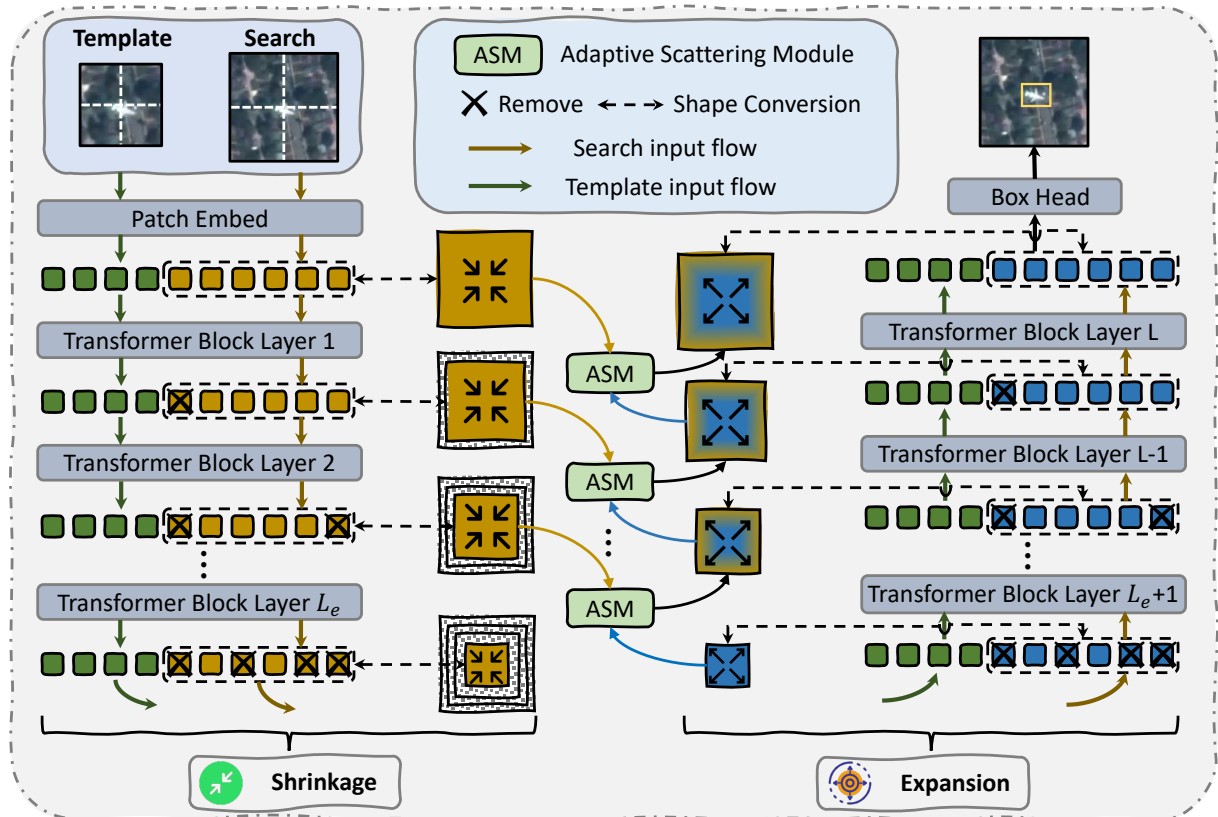

*Figure 2.* **Overview of the proposed FATrack architecture.** The framework comprises a single backbone (FA-ViT) and a prediction head. FA-ViT is structured into a shrinkage stage and an expansion stage. The Adaptive Scattering Module (ASM) is integrated into the expansion stage to guide the structured redistribution of sparse tokens, facilitating effective foreground enhancement and spatial reconstruction. FATrack achieves a favorable trade-off between speed and precision.

total input $\boldsymbol{T}^0 = [\boldsymbol{T}_Z; \boldsymbol{T}_X]$. FA-ViT adopts a symmetric encoder-decoder architecture with $L = 12$ transformer blocks, divided into $L_e = 6$ encoder layers and $L_d = 6$ decoder layers.

The encoder performs progressive spatial shrinking of the search tokens. Let $\boldsymbol{r}^{(l)}$ denote retained spatial resolution at the $l$-th layer, initialized as $\boldsymbol{r}^{(0)} = 16$ and updated by:

$$\boldsymbol{r}^{(l)} = \max(\boldsymbol{r}_{\min}, 16 - 2 \cdot l), \quad l = 1, \ldots, L_e, \quad (2)$$

where $\boldsymbol{r}_{\min} = 4$. At each layer, only the $\boldsymbol{r}^{(l)} \times \boldsymbol{r}^{(l)}$ tokens centered in the spatial grid are retained, determined by a fixed spatial indexing strategy $\mathcal{I}_l \subset \{1, \ldots, N_X\}$, while template tokens $\boldsymbol{T}_Z$ remain unchanged. The resulting token sequence becomes $\boldsymbol{T}^{(l)} = [\boldsymbol{T}_Z; \boldsymbol{T}_X[\mathcal{I}^{(l)}]] \in \mathbb{R}^{(N_Z + \boldsymbol{r}^{(l)2}) \times d}$. This strategy reduces the attention complexity from $\mathcal{O}(N^2 d)$ to $\mathcal{O}((N_Z + r_l^2)^2 d)$, with $r_l^2$ decreasing from 256 to 16, alleviating computational burden and background interference.

The decoder performs inverse token expansion, gradually restoring the token field size from $r = 4$ back to $r = 16$. At layer $l = L_e + i$, we define:

$$\boldsymbol{r}^{(l)} = \min(16, 4 + 2 \cdot i), \quad i = 1, \ldots, L_d. \quad (3)$$

To recover intermediate representations lost during shrink-

ing, we leverage cross-stage skip connections from the encoder. Specifically, we store each encoder output $\boldsymbol{T}_{\text{enc}}^{(l)}$ and retrieve it during decoder layer $L - l$. For each expansion step, we define the spatial token set as:

$$\hat{\boldsymbol{T}}_X^{(l)} = \text{ASM}\left(\boldsymbol{T}_X^{(l-1)}, \boldsymbol{T}_{\text{enc}}^{(L-l)}, \mathcal{I}^{(l)}\right), \quad (4)$$

where Adaptive Scatter Module (ASM) fills the spatially sparse decoder tokens $\boldsymbol{T}_X^{(l-1)}$ into the corresponding spatial grid defined by $\mathcal{I}^{(l)}$, and uses the encoder feature $\boldsymbol{T}_{\text{enc}}^{(L-l)}$ as guidance to complete the full spatial token map. This process ensures consistency in semantics and structure between the encoding and decoding stages, enabling detailed feature recovery without introducing redundant background information. The complete token flow across FA-ViT can be formulated as:

$$\boldsymbol{T}^0 \xrightarrow{\text{Shrink: } \boldsymbol{\mathcal{S}}^{(1 \to L_e)}} \boldsymbol{T}^{L_e} \xrightarrow{\text{Expand: } \boldsymbol{\mathcal{E}}^{(L_e+1 \to L)}} \boldsymbol{T}^L, \quad (5)$$

where $\boldsymbol{\mathcal{S}}^{(\cdot)}$ and $\boldsymbol{\mathcal{E}}^{(\cdot)}$ are the shrink and expand operations with spatial routing and ASM-enhanced reconstruction. This structured yet adaptive design enables FA-ViT to focus computational resources on meaningful foreground regions while maintaining spatial feature integrity via token recon-

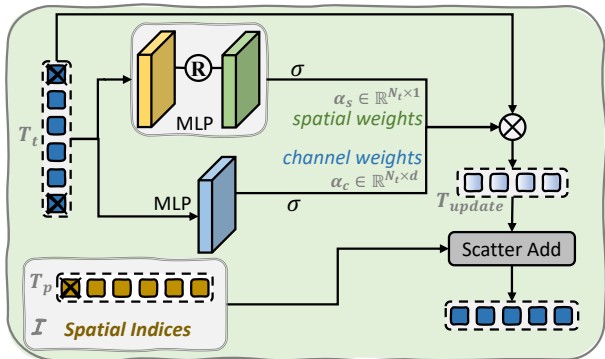

$\circledR$: Relu function  $\sigma$: Sigmoid function  $\otimes$: Element-wise multiplication

*Figure 3.* **The detailed structure of the proposed ASM.** The module receives sparse tokens together with their spatial indices and a placeholder token map, and outputs a reconstructed dense token representation. Through a combination of dual attention and scatter-add operations, it adaptively enhances and redistributes informative cues across the full token space, yielding semantically coherent and spatially continuous reconstruction.

struction. It balances the trade-off between speed and accuracy, making it better suited to satellite tracking scenarios.

### 3.3. Adaptive Scatter Module (ASM)

To facilitate the accurate reconstruction of high-resolution token representations from their sparse compressed forms in the decoding stage of FA-ViT, we introduce the Adaptive Scatter Module (ASM), as shown in Figure 3. This module is designed to model the semantic affinity between observed sparse tokens and their target spatial locations, enabling feature enhancement and structure-preserving redistribution. ASM specifically addresses the issue of information loss in unobserved regions caused by spatial token sparsification during encoding, ensuring semantic consistency and spatial continuity during reconstruction.

Let the input to ASM be a set of sparse target tokens $\boldsymbol{T}_t \in \mathbb{R}^{N_t \times d}$ (e.g., from the $4 \times 4$ token grid), a preallocated placeholder tensor $\boldsymbol{T}_p \in \mathbb{R}^{N_p \times d}$ representing the compressed token set obtained from the shrinkage stage (e.g., $8 \times 8$ or larger), which serves as the base structure for reconstructing high-resolution representations, and the corresponding spatial indices $\mathcal{I} \in \mathbb{R}^{1 \times N_t}$ indicating the target positions to be updated. ASM employs a dual-attention strategy: a spatial attention weight $\boldsymbol{\alpha}_s = \sigma(f_s(\boldsymbol{T}_t)) \in \mathbb{R}^{N_t \times 1}$, and a channel attention weight $\boldsymbol{\alpha}_c = \sigma(f_c(\boldsymbol{T}_t)) \in \mathbb{R}^{N_t \times d}$, where $f_s$ and $f_c$ are MLP-based transformations and $\sigma(\cdot)$ denotes the sigmoid function. The adaptively enhanced token representation is computed as:

$$\boldsymbol{T}_{\text{update}} = \boldsymbol{T}_t \odot \boldsymbol{\alpha}_c \odot \boldsymbol{\alpha}_s, \qquad (6)$$

where $\odot$ denotes element-wise multiplication. These refined features are then projected into the full spatial token space

using a sparse accumulation operation defined by:

$$\hat{\boldsymbol{T}}_p[:,j,:] = \sum_{i \mid \mathcal{I}_b[i]=j} \boldsymbol{T}_{\text{update}}[i], \quad \forall j \in [0, N_p), \quad (7)$$

which is efficiently implemented using "scatter and add". This operation adaptively distributes enhanced sparse tokens into a higher-resolution representation $\hat{\boldsymbol{T}}_p \in \mathbb{R}^{N_p \times d}$, guided by spatial indices $\mathcal{I}$ and modulated by learned attention. By leveraging ASM during the decoder's expansion phase, FA-ViT achieves efficient and semantically consistent token reconstruction, mitigating the drawbacks of prior foreground-centric sparsification. The module allows FA-ViT to retain critical foreground information while suppressing irrelevant background noise, supporting high-fidelity tracking performance under strict real-time constraints in satellite video scenarios.

### 3.4. Optimization

The proposed FA-ViT is optimized using a composite loss function that includes a classification loss $\mathcal{L}_{\text{CLS}}$, an bounding box regression loss $\mathcal{L}_{L_1}$, and a Generalized IoU loss $\mathcal{L}_{\text{IoU}}$ for bounding box regression. The overall objective is formulated as:

$$\mathcal{L} = \mathcal{L}_{\text{CLS}} + \lambda_{L_1} \cdot \mathcal{L}_{L_1} + \lambda_{\text{IoU}} \cdot \mathcal{L}_{\text{IoU}}, \qquad (8)$$

where $\lambda_{L_1}$ and $\lambda_{\text{IoU}}$ are regularization parameters. All losses and corresponding settings follow the (Ye et al., 2022) to ensure consistent supervision signals and fair comparison.

## 4. Experiments

### 4.1. Implementation Details

*Model:* We report two variants of our proposed FATrack, namely FATrack-Base and FATrack-Fast. The Base variant adopts the proposed FA-ViT-Base as its backbone. It is initialized with parameters pretrained via MAE (He et al., 2022), while the Fast variant employs FA-ViT-Small with random initialization. In both variants, the Adaptive Scatter Module (ASM) is inserted at each token expansion stage, with the number of ASM instances equal to the number of Transformer blocks in the decoder phase. The lightweight box head consists of a three-branch fully convolutional network, with each branch composed of four stacked Conv-BN-Relu layers, outputting four-dimensional bounding box predictions. The sizes of the template and search regions are fixed to 128×128 and 256×256, respectively. Tracking speed is measured using a single NVIDIA 2080Ti GPU.

*Training strategy:* FATrack is trained on the VISO (Yin et al., 2021) dataset's training split, supplemented by the SV248S (Li et al., 2022b) dataset with additional annotations. Cities in the SV248S test set are excluded from the

*Table 1.* **Overall performance on SV248S and SatSOT.** Highlighting the top three highest scores in red, green, and blue, respectively. Notably, the abbreviations used denote different feature extraction methods: HOG for Histogram of Oriented Gradient, CN for Color Name, CF for Convolutional Feature, and TF for Transformer.

| Method | Framework | Source | Features | SV248S | | | SatSOT | | Speed |
|---|---|---|---|---|---|---|---|---|---|
| | | | | ENUS(↑) | Succ.(↑) | Prec.(↑) | Succ.(↑) | Prec.(↑) | (*FPS*) |
| MOSSE(Bolme et al., 2010) | CF-based | CVPR2010 | GF | 0.091 | 0.078 | 0.166 | 0.269 | 0.242 | 156 |
| CSK(Henriques et al., 2012) | CF-based | ECCV2012 | GF | - | 0.050 | 0.089 | 0.247 | 0.237 | 132 |
| KCF (Henriques et al., 2014b) | CF-based | TPAMI2014 | HOG | 0.253 | 0.417 | 0.736 | 0.393 | 0.521 | 187 |
| Staple(Bertinetto et al., 2016) | CF-based | CVPR2016 | HOG | - | 0.147 | 0.343 | 0.382 | 0.462 | 38 |
| ECO (Danelljan et al., 2017) | CF-based | CVPR2017 | HOG+CN+CF | 0.236 | 0.410 | 0.731 | 0.387 | 0.549 | 240 |
| BACF(Kiani Galoogahi et al., 2017b) | CF-based | ICCV2017 | HOG | - | 0.258 | 0.696 | 0.442 | 0.550 | 19 |
| STRCF(Li et al., 2018a) | CF-based | CVPR2018 | HOG | - | 0.229 | 0.681 | 0.410 | 0.530 | 16 |
| CFME (Xuan et al., 2019) | CF-based | TGRS2019 | HOG | 0.154 | 0.284 | 0.468 | 0.428 | 0.555 | 102 |
| CPKF(Li et al., 2022a) | CF-based | TGRS2022 | HOG+CN+CF | - | - | - | 0.468 | - | 43 |
| ATOM(Danelljan et al., 2019) | DL-based | CVPR2019 | CF | 0.284 | 0.363 | 0.626 | 0.424 | 0.528 | 83 |
| SiamMask(Wang et al., 2019) | DL-based | CVPR2019 | CF | 0.147 | 0.221 | 0.565 | 0.398 | 0.552 | 110 |
| DiMP (Bhat et al., 2019) | DL-based | ICCV2019 | CF | 0.212 | 0.357 | 0.665 | 0.426 | 0.528 | 77 |
| SiamRPN++ (Li et al., 2019) | DL-based | CVPR2019 | CF | 0.213 | 0.335 | 0.663 | 0.423 | 0.537 | 56 |
| DCFST (Zheng et al., 2020) | DL-based | ECCV2020 | CF | 0.201 | 0.353 | 0.722 | - | - | 35 |
| KYS (Bhat et al., 2020) | DL-based | ECCV2020 | CF | 0.229 | 0.384 | 0.710 | - | - | 20 |
| SiamFC++(Xu et al., 2020) | DL-based | AAAI2020 | CF | 0.070 | 0.134 | 0.528 | 0.345 | 0.448 | 217 |
| Ocean (Zhang et al., 2020) | DL-based | CVPR2020 | CF | 0.084 | 0.150 | 0.414 | - | - | 25 |
| Siam R-CNN (Voigtlaender et al., 2020) | DL-based | CVPR2020 | CF | 0.127 | 0.190 | 0.328 | - | - | 5 |
| PrDiMP (Danelljan et al., 2020) | DL-based | CVPR2020 | CF | 0.195 | 0.319 | 0.585 | 0.402 | 0.465 | 47 |
| SiamCAR(Guo et al., 2020) | DL-based | CVPR2020 | CF | 0.250 | 0.448 | 0.701 | 0.446 | 0.564 | 52 |
| TransT (Chen et al., 2021b) | DL-based | CVPR2021 | CF+TF | 0.192 | 0.267 | 0.559 | 0.388 | 0.496 | 63 |
| SiamGAT (Guo et al., 2021) | DL-based | CVPR2021 | CF | 0.227 | 0.376 | 0.688 | - | - | 93 |
| STARK (Yan et al., 2021) | DL-based | ICCV2021 | CF+TF | 0.220 | 0.363 | 0.624 | 0.345 | 0.404 | 50 |
| OSTrack (Ye et al., 2022) | DL-based | ECCV2022 | TF | 0.244 | 0.399 | 0.659 | 0.359 | 0.431 | 105 |
| SeqTrack (Chen et al., 2023) | DL-based | CVPR2023 | TF | 0.272 | 0.443 | 0.705 | 0.427 | 0.512 | 40 |
| ARTrack (Wei et al., 2023) | DL-based | CVPR2023 | TF | 0.230 | 0.387 | 0.707 | - | - | 26 |
| SiamMDM(Yang et al., 2023b) | DL-based | TGRS2023 | CF | - | 0.456 | 0.725 | 0.475 | 0.611 | 62 |
| SVLPNet (Wang et al., 2024a) | DL-based | TCSVT2024 | CF+TF | 0.279 | 0.460 | 0.783 | 0.465 | 0.584 | 40 |
| ARTrackV2 (Bai et al., 2024) | DL-based | CVPR2024 | TF | 0.231 | 0.397 | 0.731 | - | - | 94 |
| ODTrack (Zheng et al., 2024) | DL-based | AAAI2024 | TF | 0.211 | 0.376 | 0.736 | - | - | 87 |
| SGLATrack (Xue et al., 2025) | DL-based | CVPR2025 | TF | 0.256 | 0.420 | 0.712 | 0.416 | 0.513 | 225 |
| ORTrack (Wu et al., 2025) | DL-based | CVPR2025 | TF | 0.251 | 0.414 | 0.706 | 0.429 | 0.534 | 226 |
| **FATrack-Fast** | **DL-based** | **-** | **TF** | **0.264** | **0.440** | **0.723** | **0.436** | **0.548** | **242** |
| **FATrack-Base** | **DL-based** | **-** | **TF** | **0.282** | **0.468** | **0.760** | **0.470** | **0.599** | **142** |

training set to prevent performance bias from data overlap. To ensure fairness, all Transformer-based baseline trackers are retrained using the same training set. During training, image pairs are sampled from sequences, and both variants use the same training pipeline to ensure consistency and comparability. Training is conducted on two NVIDIA RTX 4090 GPUs. Standard data augmentation techniques are applied (Ye et al., 2022), including random horizontal flipping and brightness jittering. The AdamW optimizer is employed with a weight decay of $1 \times 10^{-4}$, and the initial learning rate is set to $4 \times 10^{-5}$. Training proceeds for 300 epochs, each consisting of 60,000 sampled image pairs, and we decrease the learning rate by a factor of 10 after 240 epochs.

### 4.2. State-of-the-art Comparisons

**SV248S.** As shown in Table 1, FATrack-Base achieves the highest Success score (0.468) among all DL–based and CF–based trackers, demonstrating superior spatial precision and robust long-term stability. In terms of normalized error (ENUS = 0.282), it ranks second only to ATOM

(0.284) but delivers substantially higher Success and Precision, indicating stronger consistency under challenging scenarios. Compared to recent Transformer-based trackers, including SVLPNet (0.279/0.460/0.783), SGLATrack (0.256/0.420/0.712), and ODTrack (0.211/0.376/0.736), our FATrack-Base yields clearly superior accuracy while maintaining a competitive speed of 142 fps, notably faster than most ViT-based methods (e.g., TransT 63 fps, OSTrack 105 fps). The lightweight variant, FATrack-Fast, reaches an impressive 242 FPS, matching the real-time efficiency of CF-based trackers such as KCF and ECO, yet still preserving competitive accuracy (0.440 Success, 0.723 Precision). This validates the scalability and practical adaptability of our design across diverse speed–accuracy trade-offs.

Figure 4 presents attribute-based radar charts that compare the proposed FATrack with state-of-the-art methods across three metrics, ENUS, SR, and PR on the SV248S dataset. Each radar chart covers 10 challenging tracking attributes, including *Slow Motion (SM), Background Change (BCH), Continuous Occlusion (CO), and Illumination Variation*

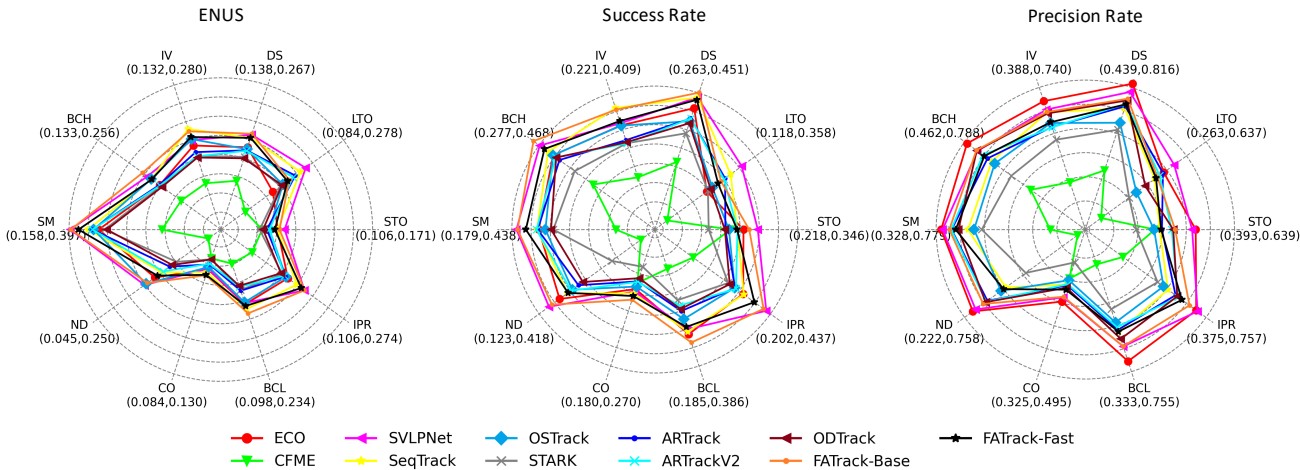

*Figure 4.* Attribute Radar Charts Comparing the Proposed FATrack with State-of-the-Art Methods Across ENUS, Success Rate (SR), and Precision Rate (PR) on the SV248S Dataset.

*Table 2.* The overall performance results on the VISO dataset are summarized, with the top three scores marked in **red**, **green**, and **blue**.

|       | KCF   | ECO   | MDNet | TransT | SiamGAT | CFME  | SiamRPN++ | DiMP  | PrDiMP | **FATrack-Fast** | **FATrack-Base** |
|-------|-------|-------|-------|--------|---------|-------|-----------|-------|--------|------------------|------------------|
| SR(↑) | 0.185 | 0.242 | 0.228 | 0.098  | 0.152   | 0.217 | 0.232     | 0.203 | 0.181  | 0.273            | 0.286            |
| PR(↑) | 0.462 | 0.607 | 0.622 | 0.384  | 0.490   | 0.547 | 0.522     | 0.583 | 0.450  | 0.615            | 0.676            |

*(IV).* The radar charts show that FATrack-Base consistently outperforms all competing methods across most attributes in all three metrics. This demonstrates our method's superior adaptability and robustness under various challenging conditions. In particular, FATrack-Base achieves notable improvements in dynamic scenes, including Illumination Variation (IV) and Dense Similarity (DS), while maintaining competitive accuracy under complex backgrounds such as BCH and ND. Meanwhile, FATrack-Fast strikes a good balance between speed and performance, ranking closely behind the base variant across most attributes. These results validate the effectiveness of the proposed tracker in handling diverse and complex satellite video scenarios, highlighting its strong generalization capability across different object appearances, motion patterns, and environmental conditions.

**SatSOT.** On this benchmark, FATrack-Base achieves the second-highest performance (SR = 0.470, PR = 0.599), surpassing strong baselines such as SVLPNet (0.465/0.584), SeqTrack (0.427/0.512), and ORTrack (0.429/0.534). These results confirm FATrack's strong discriminative capacity and robustness to scene diversity and low-texture conditions common in satellite imagery. Moreover, FATrack-Base maintains a real-time speed of 142 FPS, significantly faster than most Transformer-based trackers, while delivering higher accuracy. In contrast, traditional DCF-based methods, such as CFME (0.428/0.555), achieve lower accuracy and slower runtime (102 FPS), underscoring the efficiency gap between conventional and Transformer-driven architectures. The ultra-efficient FATrack-Fast variant further underscores the framework's deployment potential, attaining 242 FPS with minimal accuracy loss (0.440/0.548),

making it ideal for real-time onboard tracking where computational resources are limited.

**VISO.** As shown in Table 2, FATrack-Base achieves the best overall performance with an SR of 0.286 and a PR of 0.676, outperforming all previous methods. FATrack-Fast also delivers strong results (SR: 0.273, PR: 0.615), ranking second in SR and third in PR, while maintaining a lightweight configuration. Compared to state-of-the-art deep learning-based trackers such as DiMP (SR: 0.203, PR: 0.583) and SiamRPN++ (SR: 0.232, PR: 0.522), FATrack variants exhibit significant improvements in both metrics. This demonstrates the effectiveness of our proposed architecture in handling ultra-high-resolution satellite videos with dense and small-scale objects.

### 4.3. Exploration Studies

**Design choices for feature reconstruction.** Table 3(Rows #2-#4) investigates different strategies for reconstructing token representations during decoding. Replacing ASM with a simple linear mapping (#2) results in a noticeable performance drop across all datasets, particularly on VISO, indicating the importance of ASM in preserving spatial-semantic structure. Using bilinear interpolation (#3) or zero padding (#4) further degrades performance, highlighting that naive upsampling techniques are inadequate for semantically consistent token reconstruction. These results validate the design of ASM, which enhances sparse tokens and adaptively redistributes them in the expanded space, preserving both semantic alignment and structural continuity.

**Impact of Shrinkage-Expansion.** Table 3(Rows #5-#6)

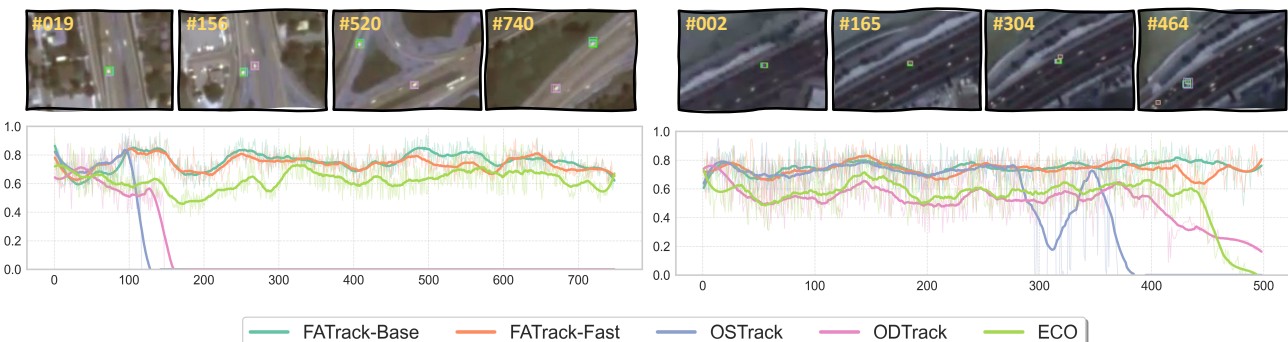

*Figure 5.* Qualitative comparisons on two representative sequences illustrating the effectiveness of the proposed FATrack. The line graphs show frame-wise Intersection over Union (IoU) scores for different trackers, while **green** bounding boxes denote the ground truth. Zooming in is recommended for detailed inspection of the tracking outputs.

*Table 3.* Exploratory evaluation of the proposed FATrack across multiple satellite video datasets, with performance measured by Success Rate (SR). Variant #6 employs an extreme compression strategy, applying the Shrinkage-Expansion process solely at the initial and final layers.

| # | Method | SV248S | SatSOT | VISO | FPS |
|---|---|---|---|---|---|
| 1 | FATrack-Base | 0.468 | 0.470 | 0.286 | 142 |
| 2 | $w/o$ ASM | 0.432 | 0.447 | 0.245 | 170 |
| 3 | $w/$ Bilinear Interpolation | 0.421 | 0.445 | 0.228 | 163 |
| 4 | $w/$ Zero Padding | 0.430 | 0.425 | 0.229 | 165 |
| 5 | $w/o$ Shrinkage-Expansion | 0.446 | 0.420 | 0.167 | 100 |
| 6 | Shrinkage-Expansion variant | 0.433 | 0.421 | 0.232 | 185 |

*Table 4.* Comparison of token filtering strategies on the SV248S dataset. All models are trained and evaluated under identical settings using an NVIDIA RTX 2080Ti GPU.

| Method | Token Filtering | MACs (G) | FPS | SR |
|---|---|---|---|---|
| OSTrack (Baseline) | Attention-Guided | 21.5 | 105 | 39.9 |
| No Filtering | - | 29.0 | 92 | 44.6 |
| Random Filtering | Random Sampling | 19.8 | 162 | 38.1 |
| Attention-based Filtering | Semantic-Guided | 21.5 | 109 | 41.5 |
| FATrack-Base | Centrality-Guided | 17.3 | 142 | 46.8 |

assess the effectiveness of the Shrinkage-Expansion mechanism in FA-ViT. Removing this mechanism entirely (#5) and using a standard ViT structure leads to the most significant performance degradation, confirming the critical role of dynamic token resolution in maintaining tracking accuracy. Variant #6, which applies an extreme compression strategy by performing shrinkage and expansion only at the initial and final layers, achieves the highest inference speed (185 fps) while maintaining moderate performance. This suggests that aggressively reducing token redundancy in intermediate layers can substantially improve efficiency, though with a slight compromise in accuracy, offering a promising trade-off for real-time applications.

**Exploration of Model Efficiency.** To further analyse the efficiency and effectiveness of the proposed Foreground-Aware Token Routing, we conducted ablation experiments comparing different token filtering strategies. As shown in Table 4, the geometry-driven Shrinkage–Expansion strategy in FATrack achieves the best balance between accuracy and computational efficiency. Specifically, FATrack attains 46.8 SR at 142 FPS with only 17.3 G MACs per

frame, representing a 40% reduction in computational cost and a substantial speed gain over No Filtering (29.0 G, 92 FPS) and attention-based filtering (21.5 G, 109 FPS), while maintaining superior accuracy. In contrast, attention-based filtering relies on dynamic attention maps, which introduce extra computation, while random filtering results in unstable localization due to the loss of spatial consistency. The geometry-driven centrality prior ensures stable spatial focus near the predicted target. It is better suited to satellite video characteristics, where motion is limited and the target remains near the center. These results confirm that performing centrality-guided pruning after patch embedding effectively removes redundant background information without sacrificing fine-grained perception, making FATrack suitable for real-time satellite tracking under limited resources.

More ablation experiments are provided in Appendix.

**Visualization.** Figure 5 demonstrates the superior robustness and stability of FATrack in challenging tracking scenarios. Across two representative sequences, both FATrack variants maintain consistently high IoU scores, outperforming existing methods. The alignment between visual outputs and IoU curves reveals FATrack's ability to preserve accurate target localization over time, highlighting its effectiveness in bridging spatial precision and temporal coherence.

## 5. Conclusion

We present FATrack, a task-driven Vision Transformer tailored for real-time satellite video tracking. FA-ViT prioritizes semantically relevant regions while maintaining computational efficiency by introducing foreground-aware token routing and a lightweight architecture. To mitigate semantic sparsity, we propose the Adaptive Scatter Module that enhances spatial continuity through selective information propagation. FATrack achieves a superior trade-off between accuracy and efficiency, outperforming both DCF- and ViT-based baselines across multiple benchmarks. This work highlights the potential of structured token routing for resource-aware visual modelling in satellite scenarios.

## Acknowledgements

This work was supported in part by the National Natural Science Foundation of China(No.62576264), Project supported by the National Science and Technology Major Project of the Ministry of Science and Technology of China (No.2025ZD0551500, No.2025ZD0551502) , the Key Project of National Natural Science Foundation of China (62431020,62231027), the Joint Fund Project of National Natural Science Foundation of China (No.U22B2054), the Fund for Foreign Scholars in University Research and Teaching Programs (the 111 Project) (No.B07048), the Postdoctoral Fellowship Program of China Postdoctoral Science Foundation (CPSF) (No.GZC20232033), the Program for Cheung Kong Scholars and Innovative Research Team in University (No.IRT 15R53), the Key Scientific Technological Innovation Research Project by Ministry of Education and the National Key Laboratory of Human-Machine Hybrid Augmented Intelligence, Xi'an Jiaotong University (No.HMHAI-202404, No. HMHAI-202405).

## Impact Statement

This work aims to advance efficient satellite video tracking for applications such as disaster response, traffic monitoring, and maritime management. Like other remote sensing technologies, it could be misused in surveillance contexts, and deployment should follow appropriate ethical and legal guidelines.

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

## A. Supplementary Material.

This section provides supplementary materials, offering additional detailed information to complement the main paper and further experimental analysis. The content in this section is organized as follows.

- **Datasets and Evaluation metrics**

- **More ablation experiments**

## B. Datasets and Evaluation metrics

**SV248S Dataset.** The **SV248S** dataset is constructed from real Jilin-1 satellite video sequences and serves as a comprehensive benchmark for small-object tracking in complex earth-observation scenes. It includes six representative video scenes (500–753 frames each) covering 248 annotated targets across multiple cities and object types such as airplanes, ships, cars, large vehicles, and other mobile platforms. Each target is categorized by difficulty level (*simple*, *normal*, or *hard*) according to size, motion, and background clutter. To provide more precise pixel-level evaluation, SV248S introduces the Normalized Union Score (NUS) and its enhanced variant ENUS, which extend IoU by considering polygonal annotations. This dataset is particularly suitable for evaluating high-precision localization and long-term stability in satellite video tracking.

**SatSOT Dataset.** The **SatSOT** dataset is the first densely annotated benchmark dedicated to single-object tracking in satellite videos. It consists of 105 sequences containing 27,664 frames across four common object categories: cars, planes, ships, and trains. Each sequence is labeled with eleven challenge attributes, including occlusion, scale variation, illumination changes, and background complexity. These attributes enable a detailed breakdown of tracker performance under various satellite-specific conditions, making SatSOT a critical benchmark for assessing robustness and generalization in this domain.

**VISO Dataset.** The **VISO** dataset is an extensive large-scale collection curated for moving object detection and tracking in high-resolution satellite videos captured by the Jilin-1 platform. It comprises 40 high resolution videos (up to $12000 \times 5000$ pixels), with more than 853,000 labeled instances across four major categories: airplanes, cars, ships, and trains. All annotations are provided as axis-aligned bounding boxes. VISO addresses key challenges in satellite vision, including extreme scale variations, low spatial resolution of moving targets, and background dynamics. It provides valuable data diversity and scale, making it essential for model pretraining and domain adaptation in satellite video understanding.

**Evaluation Metrics.** The proposed method, FATrack, is tested on three datasets using metrics like Success Rate (SR), Precision Rate (PR), and ENUS for SV248S, and SR and PR for SatSOT and VISO to evaluate the tracker's performance.

## C. More ablation experiments

*Table 5.* Comparison of feature reconstruction strategies on the SV248S dataset.

| Method | SR($\uparrow$) | PR($\uparrow$) |
|---|---|---|
| Zero Padding | 43.0 | 66.5 |
| Transposed Convolution | 41.0 | 67.8 |
| Dynamic Convolution | 43.5 | 67.2 |
| Feature Pyramid Enhancement | 44.6 | 70.9 |
| Adaptive Scatter Module (Ours) | 142 | 46.8 |

**Exploration of Feature Reconstruction Strategies.** To validate the effectiveness of the proposed Adaptive Scatter Module (ASM), we compared it with several representative feature reconstruction methods under identical configurations on the SV248S dataset. Specifically, transposed convolution, dynamic convolution, and feature pyramid enhancement were used as baselines. As shown in Table 5, although most methods outperform the zero-padding baseline, none surpass our ASM. These results demonstrate that ASM effectively restores spatial–semantic coherence under token sparsity and is particularly well-suited to the unique challenges of satellite video tracking, including low resolution and small, fine-grained targets.

**Sensitivity Analysis of Shrinkage–Expansion.** To analyze the sensitivity of the proposed FA-ViT to key hyperparameters, we further evaluated different configurations of the shrink/expand stride and minimum spatial resolution under the same training setup on the SV248S dataset. The results are shown in Table 6. The performance remains relatively stable across a

*Table 6.* Sensitivity analysis of the shrink/expand schedule on SV248S. Values denote SR / FPS.

| Stride \ Min. Spatial Resolution | 2×2 | 4×4 | 8×8 |
|---|---|---|---|
| 2 | – | **46.8 / 142** | 46.3 / 134 |
| 4 | 45.5 / 158 | 46.0 / 150 | 45.8 / 135 |

*Table 7.* Comparison of search and template cropping configurations on the SV248S dataset.

| Search Region | Search Crop Range | Template Size | Template Crop Range | SR (SV248S)↑ | FPS↑ |
|---|---|---|---|---|---|
| 192×192 | 3× | 96×96 | 1.5× | 43.1 | 155 |
| **192×192 (Ours)** | **3×** | **96×96** | **1.5×** | **45.0** | **220** |
| 192×192 | 2× | 192×192 | 2× | 30.2 | 86 |
| 160×160 | 2× | 160×160 | 2× | 33.4 | 157 |
| 144×144 | 2× | 144×144 | 2× | 31.9 | 171 |
| 128×128 | 2× | 128×128 | 2× | 32.7 | 246 |
| 256×256 | 4× | 128×128 | 2× | 44.6 | 100 |
| **256×256 (Ours)** | **4×** | **128×128** | **2×** | **46.8** | **142** |

wide range of settings, demonstrating the robustness of the proposed architecture. The manually designed configuration (stride = 2, minimum resolution = 4×4) achieves the best balance between spatial precision (SR) and real-time efficiency (FPS = 142). These findings support our design choice, which is that a moderate shrinkage ratio ensures efficient computation without degrading tracking accuracy.

**Ablation on Search–Template Cropping Configuration.** To further examine whether efficiency gains can be achieved by directly reducing the search region and template size, we conducted controlled experiments with aligned cropping ratios to ensure consistent target scale. All settings were identical across runs on the SV248S dataset. As shown in Table 7, aligning the cropping ratios between the search and template regions can improve overall efficiency but also introduces sensitivity to extreme reductions. The configuration (search 192×192, crop 3×; template 96×96, crop 1.5×) achieves a good balance between accuracy and runtime, where our method further boosts SR from 43.1 to 45.0 and FPS from 155 to 220. Similar gains are observed at higher resolutions (256×256). In contrast, overly tight cropping (e.g., 2×) severely degrades accuracy by losing contextual cues. These results confirm that the proposed token filtering mechanism achieves stable spatial focus without sacrificing contextual robustness. These findings demonstrate that while adjusting cropping ratios affects efficiency, the proposed centrality-guided token filtering consistently maintains strong accuracy–efficiency trade-offs under diverse spatial settings, validating its robustness and necessity.

*Table 8.* The planned comparison table for satellite-specific trackers. We explicitly distinguish accuracy-oriented satellite methods from efficiency-oriented / real-time methods, and annotate differences in training data, input size, runtime hardware, and representative reported results. Since the evaluation protocols are not fully aligned across methods, the penultimate column is presented only for reference rather than strict apples-to-apples comparison.

| Method | Category | Training Data | Input Size (T/S) | Runtime Hardware | SV248S/SatSOT | FPS |
|---|---|---|---|---|---|---|
| MemTrack | Accuracy-oriented satellite method | Satellite-specific setting | 192 / 384 | RTX 4090 | 54.3/57.0 | 43.79 |
| STAR | Accuracy-oriented satellite method | SatSOT-train (from SatMTB) | 128 / 256 | RTX 4090 | 52.3/53.7 | 87.3 |
| TSTrans | Accuracy-oriented satellite method | Satellite-specific setting | 128 / 256 | RTX 3090 | 40.8/53.2 | 50 |
| FATrack-Base | Efficiency-oriented / real-time method | Unified satellite training setting | 128 / 256 | RTX 2080Ti | 46.8/47.0 | 142 |
| FATrack-Fast | Efficiency-oriented / real-time method | Unified satellite training setting | 128 / 256 | RTX 2080Ti | 44.0/43.6 | 242 |

**Comparison with Satellite-Specific Trackers.** Table 8 shows a clear distinction between two groups of methods. MemTrack, STAR, and TSTrans mainly target higher tracking accuracy through richer temporal or spatiotemporal modeling, whereas FATrack-Base/Fast are designed for a stronger real-time accuracy–efficiency trade-off under a unified setting. Although the satellite-specific methods report higher scores on some benchmarks, FATrack runs substantially faster, reaching 142 FPS and 242 FPS on a 2080Ti. Moreover, on the low-power NVIDIA Jetson Xavier NX, FATrack-Base and FATrack-Fast still achieve 25 FPS and 42 FPS, respectively, further demonstrating the practical deployment potential of FATrack under constrained computing conditions. This comparison therefore highlights FATrack's main advantage: efficient real-time tracking, while making the differences in protocol, input size, and hardware explicit.

