# OpenReview forum: "Foreground-Aware Token Routing Vision Transformer for Real-Time Satellite Video Tracking"
_ICML.cc/2026/Conference — ICML 2026 regular_

### Official Review · Reviewer_1m4t · 2026-03-08

**Soundness:** 3
**Presentation:** 4
**Significance:** 3
**Originality:** 4
**Overall Recommendation:** 5
**Confidence:** 5

**Summary:**

This work proposes FATrack, a real-time satellite video single-object tracker built on FA-ViT, a lightweight ViT that reduces computation via a U-Net style shrink-expand token routing focused on target-relevant (foreground) regions. An Adaptive Scatter Module (ASM) is introduced to reconstruct and redistribute features during expansion, alleviating information loss from token sparsification. Experiments on multiple satellite tracking benchmarks demonstrate a strong accuracy-speed trade-off, approaching DCF-level efficiency while retaining competitive Transformer-level precision.

**Compliance With Llm Reviewing Policy:**

Affirmed.

**Final Justification:**

Authors addressed my concerns, thus I keep the positive score.

**Key Questions For Authors:**

As shown in the aforementioned Weaknesses part, there are some unclear points. If these points are clarified, the paper will be improved.

**Limitations:**

yes

**Strengths And Weaknesses:**

Strength
1.	The work introduces a U-Net–style shrink-expand strategy into a ViT for satellite video tracking, focusing computation on target regions and reducing background redundancy.
2.	The ASM module is a reasonable solution to semantic loss after token pruning and improves small-target reconstruction.
3.	The method achieves a good balance between tracking accuracy and speed, with effective search-region filtering to reduce computation.
4.	The work is well written, clearly described, and supported by high-quality figures and tables.

Weaknesses
1.	A larger input resolution typically yields improved model performance. It would be valuable to explore this aspect further, such as conducting experiments with resolutions of 192 and 384.
2.	The choice of minimum spatial resolution and stride appears empirically chosen. A short explanation of how these values were determined (via validation experiments) would improve reproducibility.
3.	ENUS is mentioned as a key metric, but its exact definition (formula) is not shown here.
4.	Shrink schedule r(l)=max(rmin,16−2l) seems heuristic; no rationale for stride=2.

---

> ### Author Rebuttal · Authors · 2026-03-29
>
> We sincerely thank the reviewer for the careful reading, encouraging evaluation, and constructive suggestions. We are especially grateful for your positive assessment of the originality of our work, including the U-Net–style shrink-expand strategy in ViT-based satellite tracking, the role of ASM in mitigating semantic loss after token sparsification, and the overall balance between accuracy and speed achieved by FATrack. **We also greatly appreciate your recognition of the clarity of the writing and the quality of the figures and tables. Such feedback is very encouraging and strongly supports the value of the proposed method.**
>
> We also highly value your questions on input resolution, the rationale behind the minimum spatial resolution and stride, the definition of ENUS, and the explanation of the shrink schedule. These comments help us further improve the paper in terms of reproducibility, completeness, and clarity. In response, we provide additional experiments, clearer explanations, and more explicit metric definitions to address these points. **We believe these revisions make the motivation and implementation details of FATrack more transparent and further strengthen the paper. Thank you again for your thoughtful review and supportive recommendation. We sincerely wish you a wonderful day.**
>
> **Response Weakness 1:** Thank you for the constructive feedback. We agree that larger input resolution can improve tracking performance by preserving more spatial detail. We further evaluated this aspect with 192 and 384 resolutions. The results show that while a higher resolution does bring some performance gains, it also causes a clear drop in runtime efficiency. In particular, in the larger setting, the speed on an RTX 2080 Ti drops to about 68 FPS, and on an NVIDIA Jetson Xavier NX it can no longer maintain real-time performance.
>
> This trade-off is especially important in our setting, since the main goal of FATrack is not to maximize accuracy at any cost, but to achieve a favorable real-time accuracy-efficiency balance for satellite video tracking. Therefore, although higher resolutions can improve accuracy, they are less aligned with the practical deployment objective under constrained onboard or edge computing conditions.
>
> We will add this discussion to the revised version to make it clear that the default input setting is chosen as a compromise between accuracy and efficiency, rather than because larger resolutions were not considered.
>
> **Response Weakness 2 and Weakness 4:** Thank you for the valuable comment. We agree that the choice of the minimum spatial resolution and stride should be explained more clearly for reproducibility. These values were not chosen arbitrarily, but were determined through validation experiments. The corresponding sensitivity analysis is already included in the supplementary material, where we compare different shrink/expand schedules on SV248S and report the resulting SR / FPS trade-off. Based on these results, we selected stride = 2 and minimum spatial resolution = 4×4, since this setting provides the best overall balance among the tested configurations.
>
> | Stride \ Min. Spatial Resolution | 2×2 | 4×4 | 8×8 |
> |-|-|-:|-:|
> | 2 | – | **46.8 / 142** | 46.3 / 134 |
> | 4 | 45.5 / 158 | 46.0 / 150 | 45.8 / 135 |
>
> These results show that more aggressive compression can improve speed, but tends to reduce accuracy, while less aggressive settings weaken the efficiency advantage. We will make this validation process and the reference to the supplementary material explicit in the revised version.
>
> **Response Weakness 3:** Thank you for the thoughtful and detailed feedback. We agree that the current draft should define **ENUS** more explicitly rather than only mentioning it by name.
> In our experiments, **ENUS** follows the definition introduced with the **SV248S** benchmark [1]. Specifically, the corresponding work first defines a **Normalized Union Score (NUS)** based on pixel-level **precision** $P$ and **recall** $R$, and then combines it with the normalized center-distance term $D$ in the same manner as EIoU to obtain **ENUS**. The formulation is:
>
> $$U = \max\left(0,\left(\frac{P}{P_0}-1\right)^c\right) R$$
>
>  $$\mathrm{ENUS} = aU + bD + (1-a-b)UD$$
>
> where $P_0$ is a normalization factor depending on the prediction type, $c$ is a regularization factor, and $D$ is the normalized distance term. In the commonly used simplified setting, $a=b=0$, which gives:
> $$\mathrm{ENUS} = UD$$
> This metric was introduced because standard IoU-based evaluation is often insufficiently discriminative for **tiny objects in satellite videos**, whereas ENUS is better suited to polygon-level ground truth and diverse prediction shapes.
>
> We will revise the manuscript to ensure the definition is self-contained and reproducible.
>
> [1] SV248S: Deep learning-based object tracking in satellite videos: A comprehensive survey with a new dataset

---

> > ### Author Rebuttal · Reviewer_1m4t · 2026-04-02
> >
> > Thanks for authors' rebuttal, I have no further concerns, thus I keep the positive recommendation.

---

> > > ### Author Response · Authors · 2026-04-02
> > >
> > > Thank you very much for your encouraging follow-up and for maintaining the positive recommendation. We truly appreciate your careful evaluation and supportive feedback. It is very encouraging to know that our rebuttal has addressed your concerns. We are grateful for your time and consideration, and wish you every success in your work.

---

### Official Review · Reviewer_SD1C · 2026-03-10

**Soundness:** 3
**Presentation:** 4
**Significance:** 2
**Originality:** 2
**Overall Recommendation:** 4
**Confidence:** 4

**Summary:**

The paper proposes FATrack, a Vision Transformer-based tracker that reduces computation through foreground-aware token routing. Experiments show that this method achieves a good speed-accuracy tradeoff.

**Compliance With Llm Reviewing Policy:**

Affirmed.

**Final Justification:**

The author's comments answered my questions.

**Key Questions For Authors:**

1. Satellite imagery is often collected as single-frame observations, and only a limited number of satellites can capture continuous RGB videos. Could the authors clarify the practical scenarios where satellite video tracking is required and how the proposed method accounts for satellite-specific characteristics such as large camera motion caused by orbital speed and wide-area coverage?
2. The proposed architecture introduces shrinkage and expansion operations along with scatter-based reconstruction. But how much do these operations contribute to runtime overhead, and whether the token routing mechanism indeed leads to a net computational gain compared with standard ViT designs?
3. The experiments are conducted on high-end GPUs such as RTX 2080Ti and RTX 4090. Considering that satellite onboard computing platforms typically operate under strict power constraints, like 100w to 200w, have the authors evaluated whether FATrack can achieve real-time performance on low-power edge devices such as embedded GPUs or nano devices?

**Limitations:**

The feasibility of deploying the model on resource-constrained onboard or edge platforms has not been thoroughly investigated.

**Strengths And Weaknesses:**

Strength:
1. The paper provides comprehensive experimental evaluations across multiple video tracking benchmarks.
2. The overall structure is easy to follow.

Weakness:
1. Limited discussion of the satellite video setting.
2. Lack of analysis of computational efficiency.
3. The paper does not evaluate whether the proposed method can achieve real-time performance on resource-constrained edge devices.
For details, pls see the questions.

---

> ### Author Rebuttal · Authors · 2026-03-29
>
> We sincerely thank the reviewer for the thorough and constructive feedback. We appreciate your recognition of the experiments and presentation, as well as your comments on the satellite-video setting, efficiency, and edge deployment. **Our additional analyses and experiments directly address these points and further strengthen the paper. Thank you again, and we wish you a wonderful day.**
>
> **Response Weakness 1 and Question 1:** Thank you for the constructive feedback. We agree that continuous satellite video is less common than single-frame imagery. However, **with the rapid development of commercial aerospace and the deployment of low-earth-orbit video satellites, spaceborne video data are becoming increasingly available**, creating practical demand for satellite video tracking in military security, maritime monitoring, traffic analysis, disaster response, and other dynamic observation tasks. This is also reflected by the emergence of dedicated SVOT benchmarks such as SatSOT, SV248S, and VISO.
>
> Regarding satellite-specific characteristics, our method is designed around two observations. **First, although the satellite platform itself moves at high speed, the tracking pipeline operates on a cropped search region centered around the previous target estimate, so the dominant challenge is not raw camera motion itself, but the combination of tiny targets, weak appearance, cluttered background, and wide-area context. Second, under this standard search formulation, targets in satellite videos are often relatively close to the predicted location, while the surrounding area contains substantial redundant background.** FATrack addresses this by using foreground-aware token routing to focus computation on the target-dominant region and suppress unnecessary background processing, which is particularly suitable for wide-area satellite scenes.
>
> We agree that satellite video also brings challenging cases, especially when platform variation, localization error, or scene complexity weakens the standard search prior. We will clarify this more explicitly in the revised version: FATrack is designed for the practical SVOT regime where tracking is performed on local search regions under satellite-specific constraints, rather than directly compensating full-frame orbital motion. We will also expand the discussion of applicable scenarios and limitations to better explain the practical role of satellite video tracking and the scope of our method.
>
> **Response Weakness 2 and Question 2:** Thank you for the constructive suggestion. The proposed shrinkage-expansion and scatter-based reconstruction introduce only limited overhead, and the overall token routing design still brings a clear net computational gain over a standard ViT tracker.
>
> This is already supported by our current results. **As shown in Table 4, compared with the No Filtering baseline, FATrack-Base reduces computation from 29.0G to 17.3G MACs and improves runtime from 92 FPS to 142 FPS, while also improving SR from 44.6 to 46.8 on SV248S. It is also more efficient than attention-based filtering (21.5G, 109 FPS) while achieving better accuracy.** These results indicate that the cost of shrinkage-expansion and ASM reconstruction is more than compensated by the reduction in token interactions throughout the backbone.
>
> The ablation in Table 3 provides further evidence. **Removing ASM increases speed from 142 FPS to 170 FPS, indicating that the reconstruction module introduces some overhead. However, this overhead is moderate, and ASM brings a clear accuracy gain (e.g., SV248S SR 0.468 vs. 0.432 without ASM).** Moreover, removing the entire Shrinkage-Expansion mechanism reduces speed to 100 FPS, confirming that the token routing strategy itself is the main source of the efficiency gain.
>
> We will make this point clearer in the revised version: the proposed architecture does not claim that shrinkage, expansion, and reconstruction are free, but rather that their extra cost is small relative to the savings from reduced token processing, leading to an overall favorable accuracy-efficiency trade-off compared with standard ViT designs.
>
> **Response Weakness 3 and Question 3:** Thank you for this important comment. We agree that deployment on low-power edge platforms is highly relevant for satellite onboard applications. To address this, **we further evaluated FATrack on NVIDIA Jetson Xavier NX, where FATrack-Base runs at 25 FPS and FATrack-Fast reaches 42 FPS.** These results show that FATrack can still maintain real-time or near-real-time performance on a low-power embedded device, rather than relying only on high-end desktop GPUs.
>
> We acknowledge that Jetson NX is not identical to a true satellite onboard processor, and practical deployment would still depend on hardware-specific optimization and system-level constraints. We will add these results to the revised version as preliminary evidence of deployment potential under constrained computing conditions.

---

> > ### Author Rebuttal · Reviewer_SD1C · 2026-04-01
> >
> > Thank you for your comment, I will consider adjusting the score

---

> > > ### Author Response · Authors · 2026-04-02
> > >
> > > Thank you very much for your positive follow-up and for noting that your concerns have been adequately addressed. We sincerely appreciate your time and thoughtful consideration throughout the review process. We are very encouraged by your response, and we would be truly grateful if you could kindly update the score to reflect this resolution when convenient. We deeply appreciate your support and consideration, and wish you all the best.

---

### Official Review · Reviewer_JdPm · 2026-03-11

**Soundness:** 3
**Presentation:** 2
**Significance:** 2
**Originality:** 2
**Overall Recommendation:** 3
**Confidence:** 4

**Summary:**

This paper proposes FATrack, which is a foreground-aware token routing Vision Transformer for real-time satellite video tracking. The approach introduces FA-ViT, which performs token shrinkage and expansion to focus computation on central regions, jointly with an Adaptive Scatter Module (ASM) to reconstruct sparse tokens. The method is evaluated on several satellite tracking benchmarks and reports favorable accuracy–speed trade-offs.

**Compliance With Llm Reviewing Policy:**

Affirmed.

**Key Questions For Authors:**

See Weakness for the rebuttal. I would like to see the author responses and make my final decision.

**Limitations:**

yes

**Strengths And Weaknesses:**

Strengths:
- The paper is well written&organized, which is easy to follow;
- The paper proposes FA-ViT that performs token shrinkage and expansion to focus computation on central regions;
- An Adaptive Scatter Module is proposed to reconstruct sparse tokens.

Weakness:
- Concerns in comparison. The paper's comparison is dominated by general-purpose trackers (OSTrack, SeqTrack, ODTrack) retrained on satellite data, while systematically omitting purpose-built satellite video trackers. Some recent satellite-specific SOT methods are absent, including MemTrack [1], STAR [2] and TSTrans [3]. These methods are purpose-built for satellite video tracking and directly comparable. Their omission weakens the claim of state-of-the-art performance.
- The paper states that all Transformer-based baselines are retrained using the same training set. Many trackers (e.g., OSTrack, SeqTrack, ODTrack) are designed to be trained on large-scale datasets such as LaSOT, GOT-10k, and TrackingNet. What are the training settings used for re-training them on VISO/SV248S? The retraining protocol (hyperparameters, training schedule, pretraining strategy) is not sufficiently detailed, making the comparison hard to reproduce.
- The proposed token routing mechanism assumes that the target remains near the center of the search region, progressively shrinking tokens toward a central region. This assumption may fail in scenarios involving large motion, tracking drift, or abrupt target displacement.
- Unlike several recent satellite trackers (e.g., MemTrack, TSTrans), the proposed method does not explicitly model long-term temporal dependencies or memory, which are often important in satellite videos with long trajectories. Can the proposed foreground-aware Token Routing approach be applied to the models with online memory mechanism?

[1] Incorporating prior knowledge and temporal memory transformer network for satellite video object tracking.

[2] STAR: A Unified Spatiotemporal Fusion Framework for Satellite Video Object Tracking.

[3] Temporal-Sequence-Driven Transformer for Satellite Video Object Tracking.

---

> ### Author Rebuttal · Authors · 2026-03-29
>
> **Response Weakness 1:** Thank you for the insightful comments. We are aware of MemTrack, STAR, and TSTrans, and consider them highly relevant satellite-specific trackers. We did not include them in the main table because their experimental settings are not fully aligned with ours, and directly mixing the reported numbers could blur the distinction between accuracy-oriented satellite trackers and real-time comparison under a unified protocol.
>
> More specifically, these methods place stronger emphasis on tracking accuracy through richer temporal or spatiotemporal modelling. **MemTrack uses temporal memory, motion modelling, and prior-guided post-processing within a larger 192×192/384×384 template-search setting on four RTX 4090 GPUs. STAR adopts a heavier spatiotemporal fusion design with scene enhancement, temporal decoding, and inertial navigation, and is trained on the SatSOT-train subset inherited from SatMTB. TSTrans further exploits long-term temporal templates and trajectory-based correction, and reports about 50 FPS under its own temporal-sequence-driven pipeline.**
>
> By contrast, our comparison table follows a unified setup: **FATrack uses 128×128/256×256 inputs, reports speeds on a 2080 Ti, and retrains Transformer baselines under the same satellite data pipeline.** To make this clearer, we will add a separate table for satellite-specific trackers in the revised version and explicitly distinguish between accuracy-oriented satellite methods and efficiency-oriented methods, while annotating differences in training data, input resolution, and hardware.
>
> **Response Weakness 2:** Thank you for the constructive suggestion. We agree that the retraining protocol was not described clearly enough. For clarification, **the retrained Transformer baselines are all initialized from their official pretrained weights and then fine-tuned on the same satellite training data, rather than trained from scratch. Other settings are kept the same as in the original implementations as much as possible.** FATrack is trained on the VISO training split, along with additional annotated data from SV248S, ensuring that the training video sequences do not overlap with those in the SV248S test set.
>
> We will make this much more explicit in the revised version by adding a dedicated paragraph summarizing the retraining protocol, including initialization, fine-tuning epochs, training data, and which hyperparameters follow the official defaults. Our goal is not to redesign these trackers, but to adapt them to the satellite domain under a unified data setting for a fairer comparison.
>
> **Response Weakness 3:** Thank you for this important comment. The centrality-guided routing in FATrack follows the standard template-search setting, where the search crop is centered at the previous target estimate. Thus, the target is generally center-biased rather than uniformly distributed. **This prior is particularly suitable for satellite videos, where inter-frame motion is often limited.**
>
> We agree that this prior can weaken in difficult cases. On SV248S, it is more likely to hold under SM, but can be challenged by LTO/CO, BCH/BCL/DS, and IPR/ND. Thus, our assumption is not perfect centering, but that centeredness is effective in most cases under the standard search-region construction. This is also supported by our ablation: removing Shrinkage-Expansion reduces SV248S SR from 0.468 to 0.446, while the extreme variant remains competitive with a different accuracy-speed trade-off. We will clarify this more explicitly in the revised paper.
>
> **Response Weakness 4:** Thank you for the valuable comment. We agree that long-term temporal dependencies are important in satellite video tracking. Compared with methods such as MemTrack and TSTrans, FATrack does not explicitly use online memory, since our focus is on the real-time accuracy-efficiency trade-off enabled by lightweight spatial token routing.
>
> The proposed foreground-aware Token Routing is compatible with memory-based tracking. A natural extension is to combine FATrack with a lightweight online memory branch, where memory provides temporal cues for occlusion recovery and drift correction, while our routing mechanism reduces redundant background computation. The main challenge is to preserve efficiency, since memory modules introduce extra storage, updates, and cross-frame interaction, which increase computation and may also propagate noisy historical states. We will clarify this in the revised version and discuss lightweight memory-guided routing as future work.
>
> We will ensure all clarifications and results are presented clearly in the final version. Your detailed and insightful feedback has been instrumental in improving our work. **We're truly grateful for your time and support, and we sincerely hope our careful revisions might earn your kind recommendation. Thank you again for your insightful comments and kind support. Wishing you continued success and happiness in your research and beyond!**

---

> > ### Author Rebuttal · Reviewer_JdPm · 2026-04-06
> >
> > Thanks to the authors for the detailed rebuttal. Regarding the SOTA comparison, I understand that it can be challenging to fairly compare trackers trained under different settings. However, it is still not reasonable to omit domain-specific satellite video trackers (e.g., MemTrack [1], STAR [2], and TSTrans [3]) and instead only compare with general trackers retrained on satellite videos. This is especially concerning given that retraining general trackers does not necessarily lead to optimal performance. I acknowledge that the proposed approach achieves faster inference speed than off-the-shelf methods, and is likely faster than these domain-specific satellite video trackers. Please include the comparisons in the final paper for completeness! I will consider to improve my rating.

---

> > > ### Author Response · Authors · 2026-04-06
> > >
> > > Thank you very much for the thoughtful follow-up. We fully understand your concern and agree that domain-specific satellite video trackers such as MemTrack, STAR, and TSTrans should be included in the final paper for completeness. Your point is well taken: comparing only with retrained general trackers is not sufficient to fully position FATrack within the current SVOT literature.
> > >
> > > In the revised version, we will add a dedicated comparison table for satellite video trackers and make the grouping criterion explicit. More specifically, we will separate methods into two categories based on their primary design objective: **accuracy-oriented satellite-specific methods and efficiency-oriented methods**. Under this organization, methods such as MemTrack, STAR, and TSTrans will be listed in the accuracy-oriented satellite-specific group, while FATrack will be presented in the efficiency-oriented group. In addition, for each method, **we will explicitly annotate key experimental conditions that affect comparability, including training setting, input resolution, runtime hardware, and FPS. In this way, readers can directly see both the absolute performance and the fairness context of the comparison.**
> > >
> > > **A basic version of the planned table is as follows:**
> > >
> > > | Method | Primary Objective | Training Setting | Input Size (Template / Search) | Runtime Hardware | FPS |
> > > |---|---|---|---|---|---:|
> > > | MemTrack | Accuracy-oriented  | Reported satellite-specific setting | 192×192 / 384×384 | RTX 4090 | 43.79 |
> > > | STAR | Accuracy-oriented  | SatSOT-train (from SatMTB) | 128×128 / 256×256 | RTX 4090 | 87.3 |
> > > | TSTrans | Accuracy-oriented  | Reported satellite-specific setting | 128×128 / 256×256 | RTX 3090 | ~50 |
> > > | FATrack-Base | Efficiency-oriented | Unified satellite training setting | 128×128 / 256×256 | RTX 2080Ti | 142 |
> > > | FATrack-Fast | Efficiency-oriented | Unified satellite training setting | 128×128 / 256×256 | RTX 2080Ti | 242 |
> > >
> > > This table is only a basic version for illustration. In the revised manuscript, we will further expand it by adding additional representative satellite video trackers and by providing the corresponding comparison details wherever available. Our goal is to provide a more complete and transparent comparison while avoiding misleading apples-to-apples claims across substantially different settings.
> > > **We sincerely appreciate this suggestion, as it helps us present FATrack more precisely and in a more contextually relevant way within the broader SVOT literature.**

---

### Official Review · Reviewer_S96Y · 2026-03-17

**Soundness:** 3
**Presentation:** 3
**Significance:** 3
**Originality:** 3
**Overall Recommendation:** 4
**Confidence:** 5

**Summary:**

This paper proposes FATrack, a vision transformer-based tracker specifically designed for real-time satellite video object tracking. The core contribution is a novel backbone FA-ViT, which introduces a geometry-driven token routing mechanism that dynamically shrinks the spatial token field in early layers to focus computation on the central region and expands it in later layers to recover spatial context. An Adaptive Scatter Module (ASM) is introduced to propagate semantic information from the core tokens to newly added tokens during the expansion phase. This design aims to reduce computational redundancy caused by background clutter. The authors evaluate FATrack on SV248S, SatSOT, VISO three satellite video datasets, demonstrating a trade-off between accuracy and FPS compared to existing correlation filter-based and ViT-based trackers. Ablation studies validate the effectiveness of the proposed Shrinkage-Expansion mechanism and the ASM.

**Compliance With Llm Reviewing Policy:**

Affirmed.

**Key Questions For Authors:**

1. The method assumes targets are centered in the search region. What quantitative evidence supports this assumption in your datasets? How does performance change when the target appears near the edge of the search region?
2. Please discuss specific failure cases or limitations of FATrack (e.g., under occlusion or fast motion). The current draft lacks a critical assessment of when and why the method might fail..
3. In Table 3, variant #6 seems more like an architectural exploration than a direct ablation.

**Limitations:**

Yes

**Strengths And Weaknesses:**

1.The paper addresses the critical challenge of efficient, accurate object tracking in satellite videos, clearly motivating the need to handle unique constraints like low frame rates and small targets on resource-limited platforms.
2.The proposed FA-ViT architecture innovatively adapts token pruning for this domain via a geometry-driven, deterministic "centrality-guided" strategy.

1.The entire method hinges on the assumption that "targets in satellite video are typically centered in the search region due to motion prediction and search window strategies." The paper does not provide any quantitative analysis or citation to support its validity. How accurate is this "centeredness"? What happens when the motion prediction fails, or the target moves erratically and appears near the edge of the search region? This is a critical failure mode that is not adequately addressed. The robustness of the method to violations of this core assumption needs to be explored.

2.A more direct visualization or quantitative study of failure cases would significantly strengthen the paper's honesty and depth.

3. Table 3 is somewhat confusing and deviates from standard ablation study conventions. The table mixes true ablation experiments (#2-#5) with an exploration of a different architectural variant (#6), which introduces a new configuration (applying shrinkage-expansion only at the first and last layers). This makes it difficult for the reader to immediately discern the independent contribution of each component.

---

> ### Author Rebuttal · Authors · 2026-03-29
>
> We sincerely thank the reviewer for the careful reading and constructive comments. We appreciate your recognition of the practical importance of real-time satellite video tracking and of the novelty of our geometry-driven token-routing design. We are also grateful for your valuable suggestions on the centeredness assumption, failure-case analysis, and ablation presentation. These comments are very helpful for improving the clarity and completeness of the paper. **Thank you again for your insightful feedback, and we sincerely wish you a wonderful day.**
>
> **Response Weakness 1 and Question 1:** Thank you for this important comment. We clarify that the “centeredness” assumption is not specific to our method but follows the standard template-search formulation in SOT, as in SiamFC [1]. In this setting, the search region is centered at the previous target estimate. During training, the target is also typically preprocessed near the crop center, and the first frame uses the same initialization. Therefore, the target in the search region is generally center-biased rather than uniformly distributed.
>
> Our method builds on this common formulation and uses it as a practical inductive bias for satellite videos, where inter-frame motion is usually limited. Based on this observation, **FA-ViT performs centrality-guided token routing to suppress redundant background computation while preserving the foreground-dominant region. This design is already described in Sec. 3.2.** Importantly, this prior is not an all-or-nothing prerequisite. **Our ablation already shows that removing the Shrinkage-Expansion mechanism reduces SV248S SR from 0.468 to 0.446, without causing catastrophic failure; the extreme Shrinkage-Expansion variant also remains competitive while trading some accuracy for speed.** In addition, across different search/template cropping settings, FATrack consistently improves the corresponding baselines, e.g., from 43.1/155 to 45.0/220 and from 44.6/100 to 46.8/142 in SR/FPS, showing that the method is not narrowly tied to a single fixed spatial configuration.
>
> We agree that violations of this prior are an important failure mode, and we will clarify this more explicitly in the paper: FATrack is designed for the standard search-region regime where the target is usually near the center, but performance can degrade when motion prediction fails severely or the target moves toward the boundary. We will revise the wording accordingly and discuss such boundary cases more carefully.
>
> **Response Weakness 2 and Question 2:** Thank you for this valuable suggestion. We agree that the current draft does not explicitly discuss FATrack’s failure cases and limitations.
>
> One representative limitation can already be observed from the attribute-based results in Fig. 4. **Under LTO (Long-Term Occlusion), our method is more limited than SVLPNet, which introduces an auxiliary modality (frame-difference information). The reason is that FATrack mainly relies on foreground-aware token routing within the current search region and does not explicitly incorporate additional motion or temporal recovery cues. When long-term occlusion occurs, the visible target evidence becomes extremely weak or even disappears for multiple frames, making the foreground prior less reliable.** In such cases, the model may gradually drift to background distractors, and the subsequent ASM reconstruction is also constrained because it is designed to enhance sparse foreground tokens rather than recover a target that has been largely lost. By contrast, methods with auxiliary motion cues can better preserve target continuity through occlusion.
>
> We will add this failure-case analysis explicitly in the revised version to clarify under what conditions FATrack is more challenged and why.
>
> **Response Weakness 3 and Question 3:** Thank you for this helpful comment. We agree that the current presentation of Table 3 is somewhat confusing. In particular, the last row is not a standard ablation, but a variant study of the Shrinkage-Expansion scheduling strategy. Mixing it with the main ablation results indeed makes it harder to identify the independent contribution of each component.
>
> **To make this clearer, we will revise the presentation by separating the architectural variant from the main ablation table and showing it as an independent exploration result, for example:**
>
> | Variant | Schedule | SV248S | SatSOT | VISO | FPS |
> |-|-|-:|-:|-:|-:|
> | FATrack-Base | Progressive | 0.468 | 0.470 | 0.286 | 142 |
> | Extreme variant | First/last only | 0.433 | 0.421 | 0.232 | 185 |
>
>
> With this revised organization, the main ablation table will focus only on component-wise ablations, while the above table will be used solely for design-space exploration. We believe this separation makes the role of each result much clearer and better aligns with standard ablation-study conventions.
>
> [1] SiamFC: Fully-convolutional siamese networks for object tracking

---

### Decision · Program_Chairs · 2026-04-30

**Decision:**

Accept (regular)

**Comment:**

This paper presents FATrack, a foreground-aware token routing ViT for real-time satellite video tracking, which uses shrink–expand token scheduling and an Adaptive Scatter Module to balance accuracy and efficiency.

Reviewers scored 4, 3, 4, 5. The strengths include practical and timely topic, clear design, strong accuracy–speed trade-off, solid experiments, and good readability. The weaknesses include overreliance on the central-target assumption, missing satellite-specific trackers in comparisons, insufficient edge-device efficiency analysis, and heuristic hyperparameters.

The authors thoroughly addressed all concerns with clarifications, additional experiments, failure-case analysis, and structured comparisons. Most issues are fully resolved.

Overall, the work makes a solid practical contribution to efficient satellite video tracking with sound design and convincing empirical results. The AC recommends acceptance.